# Transcriptional Landscape of Repetitive Elements in Psoriatic Skin from Large Cohort Studies: Relevance to Psoriasis Pathophysiology

**DOI:** 10.3390/ijms242316725

**Published:** 2023-11-24

**Authors:** Vidya S. Krishnan, Sulev Kõks

**Affiliations:** 1Centre for Molecular Medicine and Innovative Therapeutics, Murdoch University, Discovery Way, Murdoch, WA 1650, Australia; vidya.saraswathykrishnan@murdoch.edu.au; 2Perron Institute for Neurological and Translational Science, 8 Verdun St., Nedlands, WA 6009, Australia

**Keywords:** repetitive elements, psoriasis, HERV, tRNA, transcriptomics, LINE-1 element

## Abstract

While studies demonstrating the expression of repetitive elements (REs) in psoriatic skin using RNA-seq have been published before, not many studies have focused on the genome-wide expression patterns using larger cohorts. This study investigated the transcriptional landscape of differentially expressed REs in lesional and non-lesional skin from two previously published large datasets. We observed significant differential expression of REs in lesional psoriatic skin as well as the skin of healthy controls. Significant downregulation of several ERVs, HERVs (including HERV-K) and LINEs was observed in lesional psoriatic skin from both datasets. The upregulation of a small subset of HERV-Ks and Alus in lesional psoriatic skin was also reported. An interesting finding from this expression data was the significant upregulation and overlapping of tRNA repetitive elements in lesional and non-lesional psoriatic skin. The data from this study indicate the potential role of REs in the immunopathogenesis of psoriasis. The expression data from the two independent large study cohorts are powerful enough to confidently verify the differential expression of REs in relation to psoriatic skin pathology. Further studies are warranted to understand the functional impact of these repetitive elements in psoriasis pathogenesis, thereby expanding their significance as a potential targeting pathway for the disease treatment of psoriasis and other inflammatory diseases.

## 1. Introduction

Repetitive elements (REs) make up a significant fraction (approximately 45%) of the entire genome. They regulate the expression of a large number of genes in the human genome and consist of many different families, including endogenous retroviruses (ERVs), human endogenous retroviruses (HERVs), long and short interspersed nuclear elements (LINEs and SINEs) and SINE-VNTR-Alus (SVAs) [1,2]. An increasing number of studies have demonstrated how endogenous retroelements act as key contributors to various inflammatory and autoimmune diseases, including multiple sclerosis (MS), Aicardi–Goutières syndrome (AGS), and systemic lupus erythematosus (SLE) [2,3,4]. Psoriasis is an immune-mediated chronic inflammatory disease of the skin characterized by altered epidermal differentiation, which leads to red lesional plaques and scaling in the affected skin. Endogenous retroviral activity has been reported in psoriasis, where studies have shown [5,6] the presence of retroviral particles in the skin, urine and lymphocytes of psoriatic patients [7,8,9]. The expression of HERVs in normal human skin can be either activated or repressed using ultraviolet (UV) irradiation, which is known to clear psoriatic lesions with high efficacy [10,11]. All these earlier studies support the active role of endogenous retroelements in psoriatic lesions and in the control of keratinocyte proliferation and differentiation.

Human endogenous retroviruses (HERVs) belong to the family of long terminal repeat (LTR) retrotransposons that make up 8–10% of the human genome [12]. Previous genetic studies have shown that peripheral blood from lesional psoriatic skin and normal skin expresses the human endogenous retrovirus K (HERV-K) deoxyuridine triphosphate nucleotidohydrolase (dUTPase) transcripts, located within the psoriasis susceptibility 1 (PSOR1) locus [13]. These HERV-K dUTPase proteins induce the secretion of Th1 and THh7 cytokines involved in psoriatic plaque formation in dendritic cells and keratinocytes [14]. A large cohort case–control association study of HERV-K dUTPase variants involving 708 psoriatic and 349 healthy controls has found five common HERV-K dUTPase variants to be strongly associated with psoriasis. Furthermore, the humoral and T cell responses to HERV-K dUTPase were higher in psoriatic patients compared to that in controls [15]. However, contrary to this, Gupta and colleagues observed a significant decrease in the expression of HERV-K *env*, *gag*, *pol* and ERV-9 genes in lesional psoriatic skin as compared to that in healthy skin [16].

Non-long terminal repeat (non-LTR) retrotransposons like LINE-1 account for 17% of eukaryotic genomes [17]. They are known to trigger the type I interferon (IFN-I) pathway in auto-immune diseases, including SLE. Altered LINE-1 methylation is suggested to be one of the potential factors triggering such immune dysfunction [18]. One of the first studies to assess methylation and expression patterns of LINE-1 in psoriasis by Yooyongsatit reported hypomethylation of LINE-1 in the epidermal skin of psoriasis patients [19].

Not many studies have investigated the genome-wide expression patterns of repetitive elements in the context of psoriasis. One recent study by Lattekivi et al. in 2018 is an exception. This study analysed RNA-seq data from the skin samples of 12 psoriasis patients and 12 healthy controls and reported high levels of repetitive elements in the skin of psoriasis patients as well as that of healthy controls [12]. Interestingly, this study also reported the downregulation of differentially expressed elements from the HERV family (except HERV-K11d and HERV-K14c elements) in lesional (LP) and non-lesional (NLP) skin as previously reported by Gupta and his colleagues [16].

In the present study, we have re-analysed previously published and publicly available RNA-seq data [20,21]. Two large datasets, GSE54456 and GSE121212, derived from healthy and psoriatic patients, were downloaded from the repository. Repbase database was utilized to investigate the transcriptional landscape of differentially expressed (DE) repetitive elements (REs) in the lesional and non-lesional skin. The findings are detailed below.

## 2. Results

### 2.1. GSE54456

This dataset contains RNA sequencing data collected from 95 psoriatic and 82 normal skin samples. Comparison between psoriatic and normal skin samples resulted in 337 DE elements at FDR 0.05 (Figure 1). Out of the 337 DE elements, 43% were upregulated (146 out of 337), while the downregulated REs constituted 57% (192 out of 337). The ERVs and DNA transposons were mostly downregulated. Out of 157 differentially expressed ERVs, 110 were downregulated. The DNA transposons showed a similar profile where 17 out of 28 were downregulated. The non-LTRs, including LINEs and SINEs, were both up- and downregulated. The other group of differentially expressed RE elements included small nuclear RNAs (snRNAs), rRNAs and tRNAs. Some of the upregulated and downregulated REs are presented in Table 1. The full list of differentially expressed elements in LP and NLP groups can be found in Appendix A.

### 2.2. GSE121212

This dataset contains RNA sequencing data collected from 55 lesional skin samples (LP) and 54 non-lesional skin samples (NLP) of psoriasis patients and 38 samples from the skin of healthy controls (C).

#### 2.2.1. Differentially Expressed Repetitive Elements: LP vs. C and NLP vs. C

Comparisons between LP vs. C and NLP vs. C groups resulted in 435 and 44 differentially expressed RE elements, respectively. Compared to the respective control groups, LP expressed 49% of the upregulated genes (213 out of 435 were upregulated), and NLP expressed 86% (37 out of 43 were upregulated). In LP, the ERVs, non-LTRs (including LINEs and SINEs) and tRNA repetitive elements constituted the predominant differentially expressed Res (Figure 2). ERVs were significantly downregulated in LP samples. Out of 166 differentially expressed ERVs, 103 were downregulated. Among the non-LTRs, the LINEs (L1) were significantly downregulated in LP samples (38 out of 41 downregulated Res), and SINEs (Alu) were significantly upregulated (39 out of 48 upregulated REs). Interestingly, there was a significant upregulation of tRNA repetitive elements in LP vs. C samples. These tRNA repetitive elements were also upregulated in NLP vs. C samples (Figure 3). There was a significant overlap of differentially expressed tRNA repetitive elements (upregulated and downregulated) in the NLP group (NLP vs. C) with the LP vs. C group (Table 2). The full list of differentially expressed elements in LP and NLP groups vs. the control (C) groups can be found in Appendix A.

#### 2.2.2. Pairwise Comparison: LP vs. NLP Samples

Pairwise analysis of LP and NLP groups resulted in 329 DE elements at FDR 0.05. Out of 141 differentially expressed ERVs, 85 were downregulated. Among the non-LTRs, the SINEs (Alu) were significantly upregulated, while LINEs (L1) were significantly downregulated in LP samples. All the differentially expressed tRNA repetitive elements were upregulated in LP vs. NLP comparisons (Figure 4). The top upregulated repetitive elements in LP vs. NLP comparisons with Log2 fold change values are summarized in Table 3. The full list of differentially expressed REs can be found in Appendix A.

## 3. Discussion

Genome-wide expression of repetitive elements from RNA-seq analysis in psoriasis has been studied before; however, it was mainly confined to small cohorts. This study investigated the transcriptional landscape of repetitive elements from two large datasets: GSE54456 collected data from 95 psoriatic and 82 normal skin samples (total 177), and GSE121212 collected 55 samples from lesional skin and 54 from non-lesional skin of psoriasis patients and 38 samples from the skin of healthy controls. The findings are discussed below.

The LTRs and non-LTRs, including LINEs and SINEs, were the predominantly differentially expressed repetitive elements in this study. Similar to the results obtained by Lattekivi et al., the downregulation of LTR elements and LINEs in the LP and NLP groups suggests suppression of these elements in a pro-inflammatory environment [12]. One of the main contributing factors for this suppression of LTRs and other repetitive elements could be the epigenetic modifications in retroviral silencing by alterations in methylation, histone remodelling and RNA interference [22,23]. Whole-genome DNA methylation studies from skin lesions of psoriasis patients have reported a number of hypermethylated regions [24]. Members of highly conserved families of DNA methyl transferases (DNMTs) [22] and histone de-acetylases catalyse DNA methylation and histone modification in the human genome [25,26]. It has been shown that treatment of cells with DNMT and HDAC inhibitors can induce/reactivate endogenous retroviral activity [27]. The increased expression of DNMTs and HDACs has been reported in lesional and non-lesional psoriatic skin compared to the skin of healthy controls by Keermann et al. [12,28]. All these suggest the potential downregulation of LTR repetitive elements in the psoriatic/lesional skin samples.

Human endogenous retroviruses (HERVs) make up at least 8% of the human genome, where they are found as single or multiple copies [11]. HERVs exist as proviruses in the human genome and have been reported to play an important role in triggering antiviral immune responses in autoimmune diseases [29]. Previous studies have also reported an association between HERVs and psoriasis [7,11], and the members of the HERV-W family have been found to activate the immune system via CD14/TLR4 signalling and promote the development of a Th1 type of immune response [16,30]. More recently, Lattekivi and colleagues reported significant downregulation of HERV-K and HERV-W families in lesional and non-lesional psoriatic skin, although some subsets of HERV-Ks were upregulated [12]. In agreement with the study, we also observed the downregulation of DE elements from several HERV-K and other HERV families in lesional psoriatic skin from both datasets. This included HERV-K, HERV-K131, HERV-K11I, HERV-K14CI and HERV-K31. Previously, Gupta and colleagues also reported diminished expression of HERV-K gene transcripts and q decrease in humoral responses to HERV-K in psoriasis patients [16]. We also report an upregulation of a small subset of HERV-Ks, including HERV-K9I, HERV-K14CI, HERV-K14I, and HERV-K11DI, in lesional psoriatic skin from GSE54456 and GSE121212. The pro-inflammatory environment of the psoriatic skin seems to be the contributing factor to the suppression of HERV elements in psoriatic skin. It is known that ERV transcription can be controlled by the methylation state of genomic DNA [22]. Methylation studies have observed differentially methylated regions (DMRs) covering a large part of the genome in psoriasis skin samples [31]. Gupta and colleagues reported that the RNA degradation of HERVs at the post-transcription level could be another reason affecting HERV expression [16]. Recent studies have reported that HERV DNA/RNA accumulation is prevented by certain enzymes involved in the cytoplasmic homeostasis of nucleic acids, which could possibly provide some protection against HERV-mediated immune activation [32]. Mice deficient in Trex1 exonuclease had an accumulation of endogenous retroelements cDNA, leading to immune activation [32,33].

In addition to HERVs, we also report the significant downregulation of non-LTR LINEs in lesional psoriatic skin. LINE-1 elements can induce type 1 IFN pathways in SLE and other systemic autoimmune disorders and altered LINE-1 methylation is suggested to be one of the potential factors triggering such immune activation [18]. One of the first studies to assess the methylation levels of LINE-1 and Alu in psoriasis patients was conducted by Yooyongsatit et al. They reported LINE-1 hypomethylation in the epidermal skin of psoriasis patients along with the downregulation of genes containing LINE-1 [19]. Genome-wide RNA sequencing analysis from normal and psoriatic human skin has identified significant upregulation (17-fold) of Alu-short, dispersed element-derived siRNA in psoriasis-involved skin [34]. Interestingly, we also report the upregulation of Alu elements (39 out of 48 upregulated REs) in LP samples. Conversely, there are studies that report a lower number of edited Alu elements in lesional samples compared to the skin of healthy controls [35].

Non-transposable repetitive elements such as satellite sequences, microsatellites, and multi-copied RNA genes (including tRNA, rRNA and snRNA) have been included in the Repbase Update/Repbase—a database of TEs and other types of repeats in eukaryotic genomes [36]. An interesting finding from this study was the significant upregulation of tRNA repetitive elements in lesional and non-lesional psoriatic skin. One of the early reports of the analysis of repetitive sequences containing tRNA sequences was conducted by Lawrence and colleagues in 1985 [37]. Based on the length and cleavage site, the tRNA-derived small RNAs (tsRNAs), which constitute two classes: the stress-induced tRNA fragment, or tiRNA, which is a mature tRNA produced by a specific cut in the 28–36 nucleotide (nt) anticodon ring, and the tRNA-derived fragments (tRFs), which are 14–30 nt in length. Studies on fragments derived from tRNA (tRFs) have been of considerable interest lately, and they have been recognised as an important regulator of gene expression [38]. Aberrant tRF expression in CD4^+^ cells has been identified in SLE patients [39]. Differentially expressed tsRNAs have been studied in human hypertrophic scar fibroblasts, which participate in signalling pathways important for scar formation [40]. Not many studies have investigated the role of tRFs in the context of psoriasis. The role of epidermal isoleucyl-tRNA synthetase (IARS) was investigated in imiquimod (IMQ)-induced psoriasis-like lesions in mice. It was found that IARS expression was higher in psoriatic skin, and the application of an IARS inhibitor, mupirocin, decreased inflammatory cell infiltration in an IMQ-induced mouse model [41]. One of the first studies to identify differentially expressed (DE) tRFs from psoriatic skin lesions was conducted by Zeng et al. [42]. This study identified over 234 DE tRF transcripts in psoriatic skin compared with the skin of normal controls, among which 130 tRFs were upregulated and 104 were downregulated. The downregulation of tRF-Ile-AAT-019 in psoriatic lesions was found to have a protective role in the pathogenesis of psoriasis since it could suppress the SERPINE1 gene [42]. In our study, we found 19 DE tRNA repetitive elements overlapping with LP and NLP samples, all of which were upregulated. Interestingly, in the GSE121212 database, we have identified two DE tRNA repetitive element coding for isoleucine (tRNA-Ile-ATA and tRNA-Ile-ATT), both upregulated in lesional and non-lesional psoriatic skin. Conversely, in the GSE54456 database, all three identified differentially expressed tRNA sequences were downregulated, which also included tRNA-Ile-ATT. It is also important to note that we have not performed further analysis to confirm if the tRNA elements identified in this study are tRFs, tiRNAs or full-length repetitive sequences.

To conclude, this study describes the differential expression of repetitive elements in psoriatic and normal skin from two large cohorts. The pro-inflammatory environment seems to result in a general suppression of HERVs and non-LTR LINEs, in agreement with previous studies. Conversely, the pro-inflammatory environment also seems to upregulate the Alu elements. The interesting finding of significant upregulation of tRNA repetitive elements in lesional and non-lesional psoriatic skin needs to be further explored. The data obtained from this study add to the evidence of the potential role of repetitive elements in the immunopathogenesis of psoriasis, suggesting a complex interplay between autoimmune processes and repetitive element expression. However, these expression data alone are not sufficient to establish their roles (protective or pathological) in psoriasis. Further studies are warranted to elucidate the exact role of these repetitive elements in psoriasis pathogenesis, thereby expanding their clinical significance as a potential targeting pathway for the disease treatment of psoriasis and other inflammatory diseases.

## 4. Materials and Methods

### 4.1. Datasets Used in Our Study

RNA sequence datasets derived from healthy and psoriatic patients were downloaded from the NCBI Gene Expression Omnibus database. Sequences with accession numbers GSE54456 (PMID:24441097) [21] and GSE121212 (PMID:30641038) [20] were used for the study. GSE54456 contains data from 95 psoriatic and 84 normal skin samples (total 179), whilst GSE121212 contains 55 samples from lesional skin and 54 from non-lesional skin of psoriasis patients and 38 samples from the skin of healthy controls. GSE54456 samples were classified as healthy and lesional skin, whilst GSE121212 samples were classified as healthy, non-lesional, lesional and chronically lesioned skin. The genomic annotations were downloaded from the GENCODE database, and release 43 was used. This reference sequence corresponds to the hg38 (GRCh38.p13) human reference genome. Repetitive element (RE) annotation is based on the RepBase data from June 2022.

### 4.2. Statistical Analysis

Raw sequencing FASTQ files were used for the data analysis. Differentially expressed REs were detected by comparing psoriasis and healthy groups using Salmon TE software version 0.4 (https://pubmed.ncbi.nlm.nih.gov/29218879/, accessed 8 May 2022; https://github.com/hyunhwanjeong/, accessed 8 May 2022). Briefly based on RepBase data, the reference library was built with the “SalmonTE.py index”. The expression values from FASTQ files were called with “SalmonTE.py quant” and then used for the differential analysis with the “SalmonTE.py test” command. Differentially expressed REs were identified using *p*-Value < 0.05. R package (R version 4.3.1) *ggpubr* was then used to generate box plots to visualise the difference in expression between healthy and psoriatic samples.

## Figures and Tables

**Figure 1 ijms-24-16725-f001:**
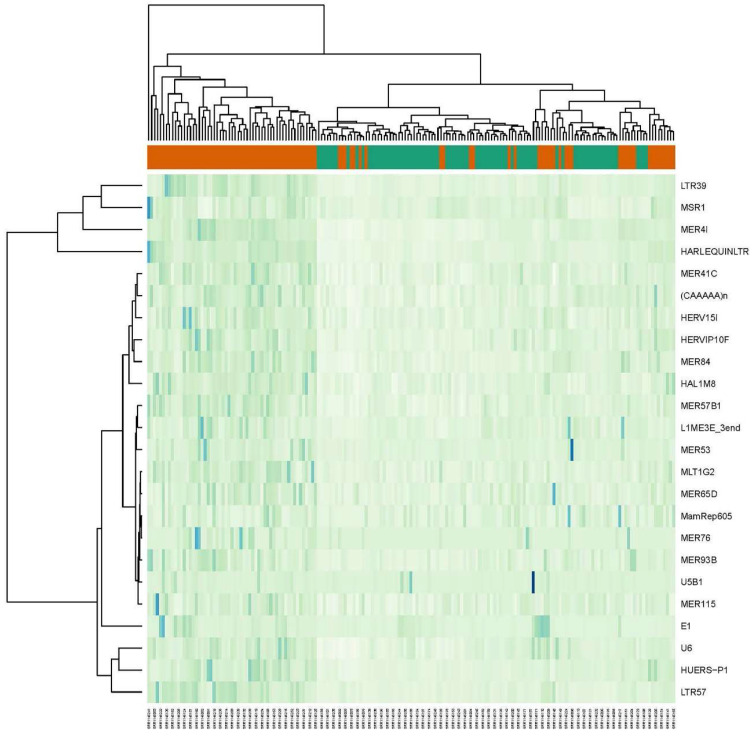
Heat map of 25 TEs with largest fold change differences between lesional psoriatic samples and healthy controls from dataset GSE54456. Orange bar is for lesional psoriatic (LP) samples and green bar is for controls (C).

**Figure 2 ijms-24-16725-f002:**
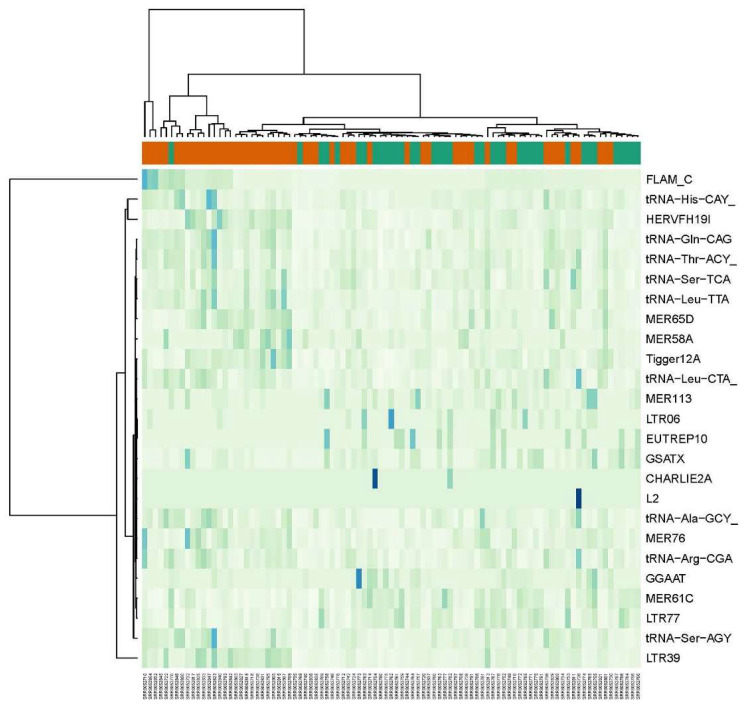
Heat map of 25 REs with largest fold change differences between lesional and healthy controls from dataset GSE121212. Orange bar is for lesional psoriatic skin (LP) samples and green bar is for control (C) samples.

**Figure 3 ijms-24-16725-f003:**
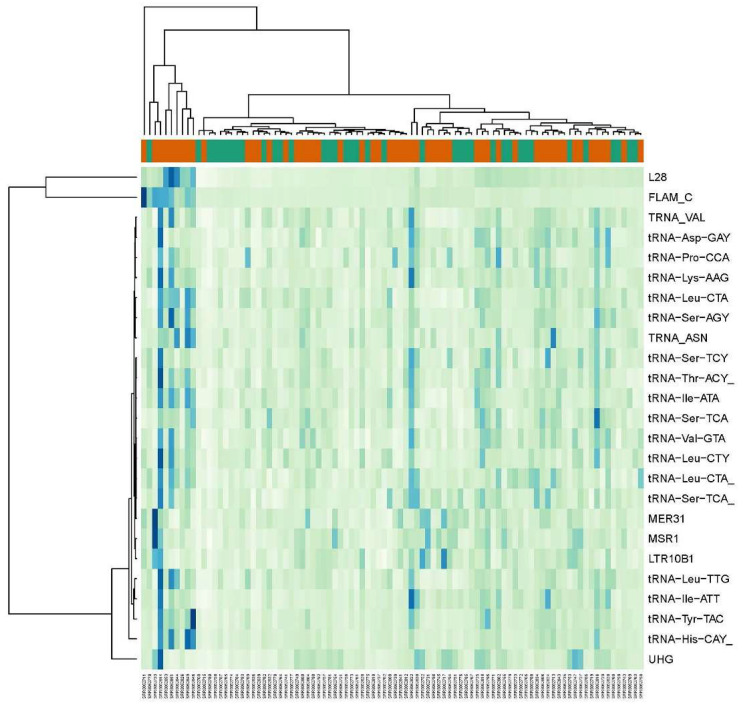
Heat map of 25 REs with largest fold change differences between non-lesional and healthy controls from dataset GSE121212. Orange bar is for non-lesional psoriatic skin (NLP) samples and green bar is for control (C) samples.

**Figure 4 ijms-24-16725-f004:**
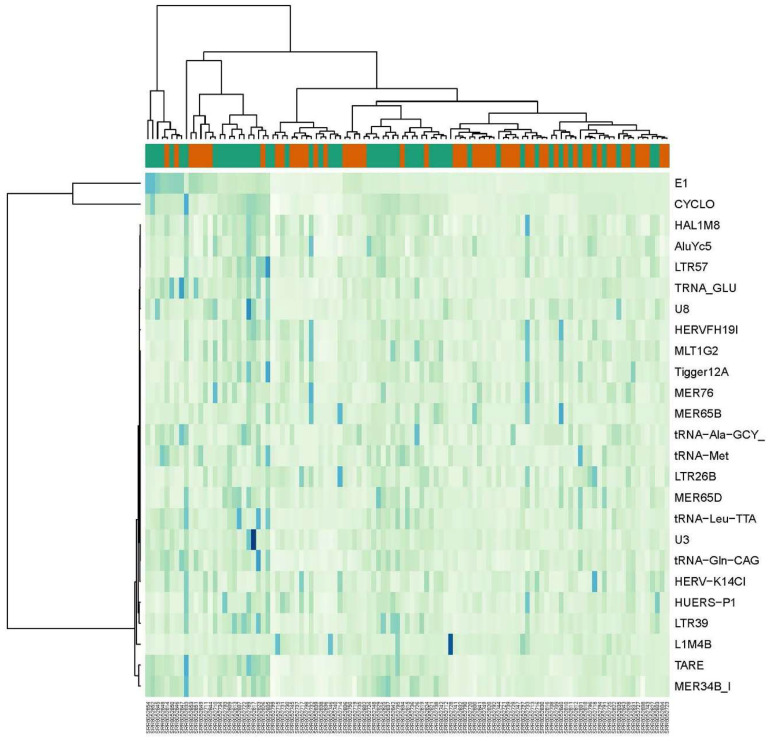
Heat map of 25 REs with largest fold change differences between lesional and non-lesional samples from dataset GSE121212. Orange bar is for lesional psoriatic skin (LP) samples and green bar is for non-lesional (NLP) samples.

**Table 1 ijms-24-16725-t001:** The top upregulated and downregulated repetitive elements in psoriatic skin among the differentially expressed elements sorted using log2FC and adjusted *p*-values. Negative log2 fold change values, which denote downregulation, are written in bold.

Element	Class	Family	log2FC	padj
LTR57	ERV3	LTR	2.156	3.15 × 10^−99^
MER53	other	other	1.764	8.16 × 10^−20^
MER93B	ERV1	LTR	1.761	7.12 × 10^−21^
MLT1G2	ERV3	LTR	1.754	2.60 × 10^−20^
E1	snRNA	snRNA	1.626	2.74 × 10^−9^
LTR39	ERV1	LTR	1.613	7.72 × 10^−65^
HUERS-P1	ERV1	LTR	1.543	1.48 × 10^−45^
HARLEQUINLTR	ERV1	LTR	1.456	2.52 × 10^−113^
MER65D	ERV1	LTR	1.391	2.80 × 10^−11^
MER4I	ERV1	LTR	1.383	3.71 × 10^−90^
MER57B1	ERV1	LTR	1.366	5.61 × 10^−32^
MER76	ERV3	LTR	1.363	0.033410871
MER76	other	other	1.363	0.033410871
MER84	ERV1	LTR	1.332	1.00 × 10^−41^
HAL1M8	L1	Non-LTR	1.270754855	1.61 × 10^−38^
**AluSx1**	**SINE**	**Non-LTR**	**−3.653**	**8.88 × 10^−9^**
**MER61C**	**ERV1**	**ERV**	**−2.068**	**9.14 × 10^−29^**
**HERV52I**	**other**	**other**	**−1.547**	**2.35 × 10^−6^**
**HERVI**	**ERV1**	**ERV**	**−1.421**	**2.64 × 10^−39^**
**LTR77**	**ERV1**	**ERV**	**−1.383**	**2.53 × 10^−27^**
**L1PA12_5**	**L1**	**Non-LTR**	**−1.325**	**9.00 × 10^−45^**
**LTR21C**	**ERV1**	**LTR**	**−1.320**	**4.53 × 10^−5^**
**LTR72B**	**ERV1**	**LTR**	**−1.312**	**4.63 × 10^−26^**
**HERV-Fc1**	**ERV1**	**LTR**	**−1.270**	**2.12 × 10^−61^**
**LTR10B2**	**ERV1**	**LTR**	**−1.249**	**9.21 × 10^−7^**
**HERVL74**	**ERV3**	**LTR**	**−1.166**	**2.22 × 10^−6^**
**(GCCCA)n**	**other**	**other**	**−1.152**	**0.030751947**
**LTR1E**	**ERV1**	**ERV**	**−1.133**	**2.27 × 10^−16^**
**CHARLIE10**	**hAT**	**DNA transposon**	**−1.001**	**0.003068318**
**HERV-Fc1_LTR1**	**ERV1**	**LTR**	**−0.999**	**6.73 × 10^−18^**

**Table 2 ijms-24-16725-t002:** Overlapping repetitive elements (upregulated and downregulated) in LP vs. C and NLP vs. C comparisons with Log2 fold change values. Negative log2 fold change values, which denote downregulation, are written in bold.

Element	Class	Family	LP vs. C	NLP vs. C
tRNA-Thr-ACY_	tRNA	tRNA	1.293066904	1.286132114
tRNA-Ser-TCA	tRNA	tRNA	1.391230455	1.159456422
FLAM_C	Non-LTR	SINE	1.451903358	1.111635811
tRNA-Ser-AGY	tRNA	tRNA	1.604957915	1.043426732
TRNA_ASN	tRNA	tRNA	1.116827011	0.990743553
tRNA-His-CAY_	tRNA	tRNA	1.328332924	0.968344648
tRNA-Pro-CCA	tRNA	tRNA	1.157233665	0.919001556
tRNA-Tyr-TAC	tRNA	tRNA	1.191472628	0.902250375
tRNA-Leu-CTY	tRNA	tRNA	1.290108017	0.852243554
tRNA-Leu-CTA	tRNA	tRNA	1.02644584	0.840100875
tRNA-Leu-CTA_	tRNA	tRNA	1.298613682	0.837939122
tRNA-Val-GTA	tRNA	tRNA	1.161453322	0.834499718
tRNA-Ser-TCA_	tRNA	tRNA	0.690630021	0.82749649
tRNA-Leu-TTG	tRNA	tRNA	1.244496472	0.798066989
TRNA_VAL	tRNA	tRNA	1.172542094	0.722428146
tRNA-Ser-TCY	tRNA	tRNA	0.824762217	0.718736735
tRNA-Ile-ATA	tRNA	tRNA	0.746887007	0.656307557
tRNA-Ile-ATT	tRNA	tRNA	0.660847114	0.627303069
tRNA-Asp-GAY	tRNA	tRNA	0.703518619	0.569808645
tRNA-Lys-AAG	tRNA	tRNA	1.151490313	0.56568494
L28	other	other	1.087847047	0.470932855
LTR10B1	LTR	ERV	0.94710745	0.454790786
UHG	snRNA	snRNA	1.000824342	0.454790786
MSR1	MSAT	Satellite DNA	0.867625814	0.441008334
LTR39	LTR	ERV1	1.678442238	0.384521163
HERVFH19I	other	other	1.37129888	0.316848192
LTR22A	LTR	ERV2	0.421760889	0.286088783
SVA_D	other	other	0.47558032	0.221054458
LTR7Y	LTR	ERV3	0.177098265	0.201687074
SVA_E	other	other	0.284551435	0.187221514
MIR	Non-LTR	SINE	0.484129605	0.178523372
LTR7B	LTR	ERV3	0.19895453	0.173423136
LTR5_Hs	LTR	ERV2	0.16541693	0.154152109
**TIGGER1**	**Mariner/Tc1**	**DNA transposon**	**−0.129994204**	**−0.084887324**
**MER57A_I**	**other**	**other**	**−0.184037934**	**−0.168413678**

**Table 3 ijms-24-16725-t003:** The top upregulated repetitive elements in LP vs. NLP comparisons with log2 fold change values. Negative log2 fold change values, which denote downregulation, are written in bold.

Element	Class	Family	log2FC	padj
tRNA-Leu-TTA	tRNA	tRNA	2.045392601	2.04482 × 10^−10^
MER65D	LTR	ERV1	1.953056271	4.34672 × 10^−17^
MER65B	LTR	ERV1	1.453383936	0.002831687
tRNA-Met	tRNA	tRNA	1.32433835	0.000874453
Tigger12A	Mariner/Tc1	DNA transposon	1.310202088	3.54274 × 10^−11^
LTR39	LTR	ERV1	1.290613523	7.5422 × 10^−18^
tRNA-Ala-GCY_	tRNA	tRNA	1.12708805	0.008585785
HERVFH19I	other	other	1.051720902	1.58521 × 10^−21^
LTR57	LTR	ERV3	1.049730212	6.56032 × 10^−13^
tRNA-Gln-CAG	tRNA	tRNA	0.99476115	8.84789 × 10^−5^
**LTR77**	**LTR**	**ERV1**	**−1.426539627**	**7.03882 × 10^−20^**
**MER61C**	**LTR**	**ERV1**	**−1.376418897**	**2.98499 × 10^−17^**
**MER113**	**hAT**	**DNA transposon**	**−1.290109724**	**0.015784932**
**HSAT5**	**Satellite**	**Satellite**	**−1.270343041**	**0.010678003**
**GSATX**	**Satellite**	**Satellite**	**−1.258391507**	**0.007646945**
**MLT1HI**	**LTR**	**ERV3**	**−1.002012749**	**0.012945334**
**HERV-Fc1**	**LTR**	**ERV1**	**−0.981094421**	**6.71427 × 10^−20^**
**LTR10B2**	**LTR**	**ERV1**	**−0.916546193**	**6.33106 × 10^−6^**
**LTR72B**	**LTR**	**ERV1**	**−0.898530987**	**1.67584 × 10^−6^**
**CHARLIE10**	**hAT**	**DNA transposon**	**−0.880609909**	**0.002559105**

## Data Availability

RNA sequence datasets derived from publicly available data of healthy and psoriatic patients downloaded from the NCBI Gene Expression Omnibus database. Sequences with accession number GSE54456 (PMID:24441097) [21] and GSE121212 (PMID:30641038) [20] were used for the study.

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
