# Peer review of "Transcriptional Landscape of Repetitive Elements in Psoriatic Skin from Large Cohort Studies: Relevance to Psoriasis Pathophysiology"

_ijms, 2023, doi:10.3390/ijms242316725_

Round 1

Reviewer 1 Report

Comments and Suggestions for Authors

The content of this paper was to report the study of the transcriptional landscape of repetitive elements in psoriatic skin from large cohort studies.

They provide useful information and some novel ideas. However, the section of the Materials and Methods needs to be enhanced. Some criteria need to be illustrated.

1.     In “Table 1. The top upregulated and downregulated repetitive elements in psoriatic skin amongst the differentially expressed elements sorted by log2 FC and adjusted p-values.”

Please describe the reason for using the log2 FC and adjusted p-values

2.     In Table, what is the meaning of LP vs C and NLP vs 129 C?

3.     It is inadequate to perform the pairwise comparison of LP vs NLP for so much data. Please consult with a statistician.

Comments on the Quality of English Language

Minor editing of English language required

Author Response

The content of this paper was to report the study of the transcriptional landscape of repetitive elements in psoriatic skin from large cohort studies. They provide useful information and some novel ideas. However, the section of the Materials and Methods needs to be enhanced. Some criteria need to be illustrated.

  1. In “Table 1. The top upregulated and downregulated repetitive elements in psoriatic skin amongst the differentially expressed elements sorted by log2 FC and adjusted p-values.” Please describe the reason for using the log2 FC and adjusted p-values

The log2FC represents the magnitude of change in gene expression between two conditions. This feature ensures that a 2-fold change up is conceptually equivalent in magnitude to a 2-fold change down. Moreover, to control the rate of false positive results, we applied adjusted/corrected p-values in the case of multiple testing. Adjusted p-values represent statistical significance between the identified differences in gene expression between samples, and this approach also enhances the reliability of our findings.

  1. In Table, what is the meaning of LP vs C and NLP vs 129 C?

Table 2 (Page 6) shows overlapping repetitive elements (upregulated and downregulated) in LP (Lesional psoriatic skin) vs C(controls) and NLP (non-lesional psoriatic skin) vs C (controls) comparisons with Log2 fold change values. 129 is the line number in the manuscript. The reviewer has mistakenly read it as NLP vs 129 C.

  1. It is inadequate to perform the pairwise comparison of LP vs NLP for so much data. Please consult with a statistician.

I am a statistician and by pairwise comparison we meant comparison between two different study groups, LP and NLP. Pairwise comparison does not have anything to do with sample size. On the contrary, a larger sample size increases the confidence and reliability of the pairwise comparisons.

Reviewer 2 Report

Comments and Suggestions for Authors

The authors review the role of repetitive elements in psoriasis. The paper is interesting and well-written. There are some issues that need to be addressed. 

1) The authors should address the issue of whether differential expression of the various repetitive elements have clinical relevance. I realize there are multiple changes, but which changes cause the inflammatory changes leading to skin lesions and arthritis? Is it feasible to determine which changes cause phenotypic changes? Maybe a discussion about this issue is warranted. 

2) You examine blood changes from one study, and lesional changes from another, are there similar or different changes when comparing to psoriatic blood versus skin?  Or are there similar changes that might focus the more important roles of repetitive elements?

3) You should add more labels or legends on your figures. You have many lines on the X and Y axes that have no explanation. What do those lines that look like family relationships mean? 

4) On line 32, you use an abbreviation (SVAs) that needs to be written out. 

5) At the end of the paper, you just copy the editor's requests for informed consent, data availability, and acknowledgements. You should add your own statement about consent, and if there are no data availability statement and acknowledgements, then you should state so. Do not include the editor's instructions. 

Author Response

The authors review the role of repetitive elements in psoriasis. The paper is interesting and well-written. There are some issues that need to be addressed.

1) The authors should address the issue of whether differential expression of the various repetitive elements have clinical relevance. I realize there are multiple changes, but which changes cause the inflammatory changes leading to skin lesions and arthritis? Is it feasible to determine which changes cause phenotypic changes? Maybe a discussion about this issue is warranted.

As mentioned in the concluding paragraph of the discussion section, this data obtained from this study add to the evidence of the potential role of repetitive elements in psoriasis development. However, this cannot be determined from expression data alone, but calls for further research to expand its clinical significance. The following text has now been added to the discussion.

To conclude, this study describes differential expression of repetitive elements in psoriatic and normal skin from two large cohorts. The pro-inflammatory environment seems to result in a general suppression of HERVs and non-LTR LINEs in agreement with previous studies. Conversely the proinflammatory environment also seems to upregulate the Alu elements. The interesting finding of significant upregulation of t-RNA repetitive elements in lesional and non-lesional psoriatic skin needs to be further explored. The data obtained from this study adds to the evidence of the potential role of repetitive elements in the immunopathogenesis of psoriasis suggesting a complex interplay between autoimmune processes and repetitive element expression. However, this expression data alone is not sufficient to establish their roles (protective or pathological) in psoriasis. Further studies are warranted to elucidate the exact role these repetitive elements might play in psoriasis pathogenesis thereby expanding its clinical significance as a potential targeting pathway for the disease treatment, for psoriasis and other inflammatory diseases.

2) You examine blood changes from one study, and lesional changes from another, are there similar or different changes when comparing to psoriatic blood versus skin?  Or are there similar changes that might focus the more important roles of repetitive elements?

This study focused on the expression of repetitive elements from skin samples of lesional psoriatic (LP), non-lesional psoriatic (NLP) psoriatic patients and healthy controls from two large datasets. We have not studied/compared blood transcriptomic profile from these datasets.

3) You should add more labels or legends on your figures. You have many lines on the X and Y axes that have no explanation. What do those lines that look like family relationships mean?

The lines on the heatmap are dendrograms which represents hierarchical clustering calculation. The row dendrograms depicts clustering of the repetitive elements and the column dendrograms depicts lesional (LP), non-lesional (NLP) or control samples.  

4) On line 32, you use an abbreviation (SVAs) that needs to be written out.

Thank you for pointing the error. We have expanded the abbreviation in the manuscript (highlighted in yellow with track changes)

5) At the end of the paper, you just copy the editor's requests for informed consent, data availability, and acknowledgements. You should add your own statement about consent, and if there are no data availability statement and acknowledgements, then you should state so. Do not include the editor's instructions.

Apologies. This was a formatting error while changing the manuscript in word document to the journal template. We have now added statements for consent, data availability and acknowledgments.